# Optimal multisensory decision-making in a reaction-time task

**Jan Drugowitsch[1,2,4]\*, Gregory C DeAngelis[1†], Eliana M Klier[3], Dora E Angelaki[3†], Alexandre Pouget[1,4†]**

[1]Department of Brain and Cognitive Sciences, University of Rochester, New York, United States; [2]Institut National de la Santé et de la Recherche Médicale, École Normale Supérieure, Paris, France; [3]Department of Neuroscience, Baylor College of Medicine, Houston, United States; [4]Département des Neurosciences Fondamentales, Université de Genève, Geneva, Switzerland

**Abstract** Humans and animals can integrate sensory evidence from various sources to make decisions in a statistically near-optimal manner, provided that the stimulus presentation time is fixed across trials. Little is known about whether optimality is preserved when subjects can choose when to make a decision (reaction-time task), nor when sensory inputs have time-varying reliability. Using a reaction-time version of a visual/vestibular heading discrimination task, we show that behavior is clearly sub-optimal when quantified with traditional optimality metrics that ignore reaction times. We created a computational model that accumulates evidence optimally across both cues and time, and trades off accuracy with decision speed. This model quantitatively explains subjects's choices and reaction times, supporting the hypothesis that subjects do, in fact, accumulate evidence optimally over time and across sensory modalities, even when the reaction time is under the subject's control.

\*For correspondence: jdrugo@gmail.com

†These authors contributed equally to this work

## Introduction

Effective decision making in an uncertain, rapidly changing environment requires optimal use of all information available to the decision-maker. Numerous previous studies have examined how integrating multiple sensory cues—either within or across sensory modalities—alters perceptual sensitivity (*van Beers et al., 1996*; *Ernst and Banks, 2002*; *Battaglia et al., 2003*; *Fetsch et al., 2009*). These studies generally reveal that subjects' ability to discriminate among stimuli improves when multiple sensory cues are available, such as visual and tactile (*van Beers et al., 1996*; *Ernst and Banks, 2002*), visual and auditory (*Battaglia et al., 2003*), or visual and vestibular (*Fetsch et al., 2009*) cues. The performance gains associated with cue integration are generally well predicted by models that combine information across senses in a statistically optimal manner (*Clark and Yuille, 1990*). Specifically, we consider cue integration to be optimal if the information in the combined, multisensory condition is the sum of that available from the separate cues (see *Supplementary file 1* for formal statement) (*Clark and Yuille, 1990*).

Previous studies and models share a common fundamental limitation: they only consider situations in which the stimulus duration is fixed and subjects are required to withhold their response until the stimulus epoch expires. In natural settings, by contrast, subjects usually choose for themselves when they have gathered enough information to make a decision. In such contexts, it is possible that subjects integrate multiple cues to gain speed or to increase their proportion of correct responses (or some combination of effects), and it is unknown whether standard criteria for optimal cue integration apply. Indeed, using a reaction-time version of a multimodal heading discrimination task, we demonstrate here that human performance is markedly suboptimal when evaluated with standard criteria that ignore reaction times. Thus, the conventional framework for optimal cue integration is not applicable to behaviors in which decision times are under subjects' control.

**eLife digest** Imagine trying out a new roller-coaster ride and doing your best to figure out if you are being hurled to the left or to the right. You might think that this task would be easier if your eyes were open because you could rely on information from your eyes and also from the vestibular system in your ears. This is also what cue combination theory says—our ability to discriminate between two potential outcomes is enhanced when we can draw on more than one of the senses.

However, previous tests of cue combination theory have been limited in that test subjects have been asked to respond after receiving information for a fixed period of time whereas, in real life, we tend to make a decision as soon as we have gathered sufficient information. Now, using data collected from seven human subjects in a simulator, Drugowitsch et al. have confirmed that test subjects do indeed give more correct answers in more realistic conditions when they have two sources of information to rely on, rather than only one.

What makes this result surprising? Traditional cue combination theories do not consider that slower decisions allow us to process more information and therefore tend to be more accurate. Drugowitsch et al. show that this shortcoming causes such theories to conclude that multiple information sources might lead to worse decisions. For example, some of their test subjects made less accurate choices when they were presented with both visual and vestibular information, compared to when only visual information was available, because they made these choices very rapidly.

By developing a theory that takes into account both reaction times and choice accuracy, Drugowitsch et al. were able to show that, despite different trade-offs between speed and accuracy, test subjects still combined the information from their eyes and ears in a way that was close to ideal. As such the work offers a more thorough account of human decision making.

On the other hand, there is a large body of empirical studies that has focused on how multisensory integration affects reaction times, but these studies have generally ignored effects on perceptual sensitivity (*Colonius and Arndt, 2001*; *Otto and Mamassian, 2012*). Some of these studies have reported that reaction times for multisensory stimuli are faster than predicted by 'parallel race' models (*Raab, 1962*; *Miller, 1982*), suggesting that multisensory inputs are combined into a common representation. However, other groups have failed to replicate these findings (*Corneil et al., 2002*; *Whitchurch and Takahashi, 2006*) and it is unclear whether the sensory inputs are combined optimally. Thus, multisensory integration in reaction time experiments remains poorly understood, and there is no coherent framework for evaluating optimal decision making that incorporates both perceptual sensitivity and reaction times. We address this substantial gap in knowledge both theoretically and experimentally.

For tasks based on information from a single sensory modality, diffusion models (DMs) have proven to be very effective at characterizing both the speed and accuracy of perceptual decisions, as well as speed/accuracy trade-offs (*Ratcliff, 1978*; *Ratcliff and Smith, 2004*; *Palmer et al., 2005*) (where accuracy is used in the sense of percentage of correct responses). Here, we develop a novel form of DM that not only integrates evidence optimally over time but also across different sensory cues, providing an optimal decision model for multisensory integration in a reaction-time context. The model is capable of combining cues optimally even when the reliability of each sensory input varies as a function of time. We show that this model reproduces human subjects' behavior very well, thus demonstrating that subjects near-optimally combine momentary evidence across sensory modalities. The model also predicts the counterintuitive finding that discrimination thresholds are often increased during cue combination, and demonstrates that this departure from standard cue-integration theory is due to a speed-accuracy tradeoff.

Overall, our findings provide a framework for extending cue-integration research to more natural contexts in which decision times are unconstrained and sensory cues vary substantially over time.

## Results

We collected behavioral data from seven human subjects, A–G, performing a reaction-time version of a heading discrimination task (*Gu et al., 2007*, *2008*, *2010*; *Fetsch et al., 2009*) based on optic flow

alone (visual condition), inertial motion alone (vestibular condition), or a combination of both cues (combined condition, *Figure 1A*). In each stimulus condition, the subjects experienced forward translation with a small leftward or rightward deviation, and their task was to report whether they moved leftward or rightward relative to (an internal standard of) straight ahead (*Figure 1B*). In the combined condition, visual and vestibular cues were always spatially congruent, and followed temporally synchronized Gaussian velocity profiles (*Figure 1C*). Reliability of the visual cue was varied randomly across trials by changing the motion coherence of the optic flow stimulus (three coherence levels). For subjects B, D, and F, an additional experiment with six coherence levels was performed (denoted as B2, D2, F2). In contrast to previous tasks conducted with the same apparatus (*Fetsch et al., 2009*; *Gu et al., 2010*), subjects did not have to wait until the end of the stimulus presentation, but were allowed to respond at any time throughout the trial, which lasted up to 2 s.

For all conditions and all subjects, heading discrimination performance improved with an increase in heading direction away from straight ahead and with increased visual motion coherence. Let $h$ denote the heading angle relative to straight ahead ($h > 0$ for right, $h < 0$ for left), and $|h|$ its magnitude. Larger values of $|h|$ simplified the discrimination task, as reflected by a larger fraction of correct choices (*Figure 2A* for subject D2, *Figure 3—figure supplement 1* for other subjects). To quantify discrimination performance, we fitted a cumulative Gaussian function to the psychometric curve for each stimulus condition and coherence. A lower discrimination threshold, given by the standard deviation of the fitted Gaussian, indicates a steeper psychometric curve and thus better performance. For both the visual and combined conditions, discrimination thresholds consistently decreased with an increase in motion coherence (*Figure 2B* for subject D2, *Figure 2—figure supplement 1* for other subjects), indicating that increasing coherence improves heading discrimination.

## Sub-optimal cue combination?

Traditional cue combination models predict that the discrimination threshold in the combined condition should be smaller than that of either unimodal condition (*Clark and Yuille, 1990*). With a fixed stimulus duration, this prediction has been shown to hold for visual/vestibular heading discrimination in both human and animal subjects (*Fetsch et al., 2009*, *2011*), consistent with optimal cue combination. In contrast, the discrimination thresholds of subjects in our reaction-time task appear to be substantially sub-optimal. For the example subject of *Figure 2A*, psychometric functions in the combined condition lie between the visual and vestibular functions. Correspondingly, discrimination thresholds for the combined condition are intermediate between visual and vestibular thresholds for this subject, and for high coherences, are substantially greater than the optimal predictions (*Figure 2B*).

This pattern of results was consistent across subjects (*Figure 2C*, *Figure 2—figure supplement 1*). In no case did subjects feature a significantly lower discrimination threshold in the combined condition than the better of the two unimodal conditions (p>0.57, one-tailed, *Supplementary file 2A*). For the largest visual motion coherence (70%), all subjects except one showed thresholds in the combined condition that were significantly greater than visual thresholds and significant greater than optimal predictions of a conventional cue-integration scheme (p<0.05, *Supplementary file 2A*). These data lie in stark contrast to previous reports using fixed duration stimuli (*Fetsch et al., 2009*, *2011*) in which combined thresholds were generally found to improve compared to

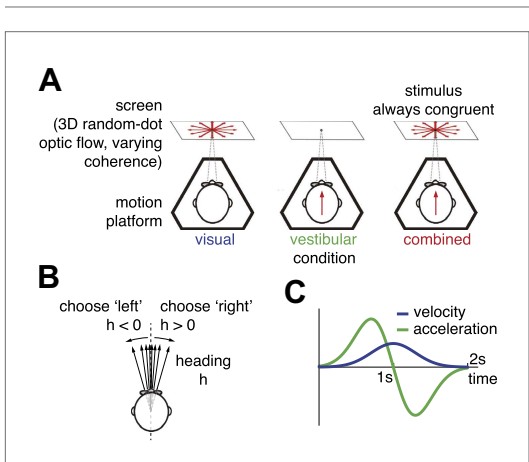

**Figure 1**. Heading discrimination task. (**A**) Subjects are seated on a motion platform in front of a screen displaying 3D optic flow. They perform a heading discrimination task based on optic flow (visual condition), platform motion (vestibular condition), or both cues in combination (combined condition). Coherence of the optic flow is constant within a trial but varies randomly across trials. (**B**) The subjects' task is to indicate whether they are moving rightward or leftward relative to straight ahead. Both motion direction (sign of $h$) and heading angle (magnitude of $|h|$) are chosen randomly between trials. (**C**) The velocity profile is Gaussian with peak velocity ~1 s after stimulus onset.

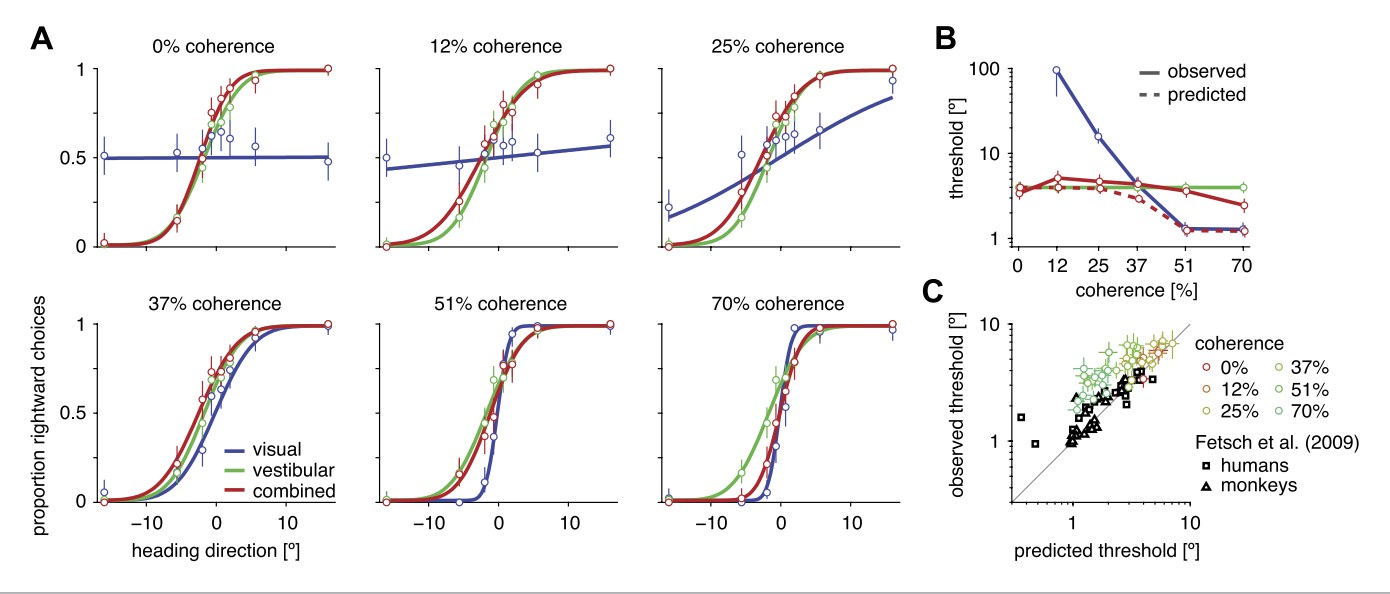

**Figure 2**. Heading discrimination performance. (**A**) Plots show the proportion of rightward choices for each heading and stimulus condition. Data are shown for subject D2, who was tested with 6 coherence levels. Error bars indicate 95% confidence intervals. (**B**) Discrimination threshold for each coherence and condition for subject D2 (see *Figure 2—figure supplement 1* for discrimination thresholds of all subjects). For large coherences, the threshold in the combined condition (solid red curve) lies between that of the vestibular and visual conditions, a marked deviation from the standard prediction (dashed red curve) of optimal cue integration theory. (**C**) Observed vs predicted discrimination thresholds for the combined condition for all subjects. Data are color coded by motion coherence. Error bars indicate 95% CIs. For most subjects, observed thresholds are significantly greater than predicted, especially for coherences greater than 25%. For comparison, analogous data from monkeys and humans (black triangles and squares, respectively) are shown from a previous study involving a fixed-duration version of the same task (*Fetsch et al., 2009*).

The following figure supplements are available for figure 2:

**Figure supplement 1**. Discrimination thresholds for all subjects and conditions.

the unimodal conditions, as expected by standard optimal multisensory integration models. To summarize this contrast, we compare the ratio of observed to predicted thresholds in the combined condition for our subjects to human and monkey subjects performing a similar task in a fixed duration setting (*Fetsch et al., 2009*). We found this ratio to be significantly greater for our subjects (*Figure 2C*; two-sample *t* test, t (77) = 3.245, p=0.0017). This indicates that, with respect to predictions of standard multisensory integration models, our subjects performed significantly worse than those engaged in a similar fixed-duration task.

A different picture emerges if we take not only discrimination thresholds but also reaction times into account. Short reaction times imply that subjects gather less information to make a decision, yielding greater discrimination thresholds. Longer reaction times may decrease thresholds, but at the cost of time. Consequently, if subjects decide more rapidly in the combined condition than the visual condition, they might feature higher discrimination thresholds in the combined condition even if they make optimal use of all available information. Thus, to assess if subjects perform optimal cue combination, we need to account for the timing of their decisions.

Average reaction times depended on stimulus condition, motion coherence, and heading direction. In general, reaction times were faster for larger heading magnitudes, and reaction times in the vestibular condition were faster than those in the visual condition (*Figure 3* for subject D2, *Figure 3—figure supplement 1* for other subjects). In the combined condition, however, reaction times were much shorter than those seen for the visual condition and were comparable to those of the vestibular condition (*Figure 3*). Thus, subjects spent substantially more time integrating evidence in the visual condition, which boosted their discrimination performance when compared to the combined condition. Note also that discrimination thresholds in the combined condition were substantially smaller than vestibular thresholds, especially at 70% coherence (*Figures 2 and 3*). Thus, adding optic flow to a

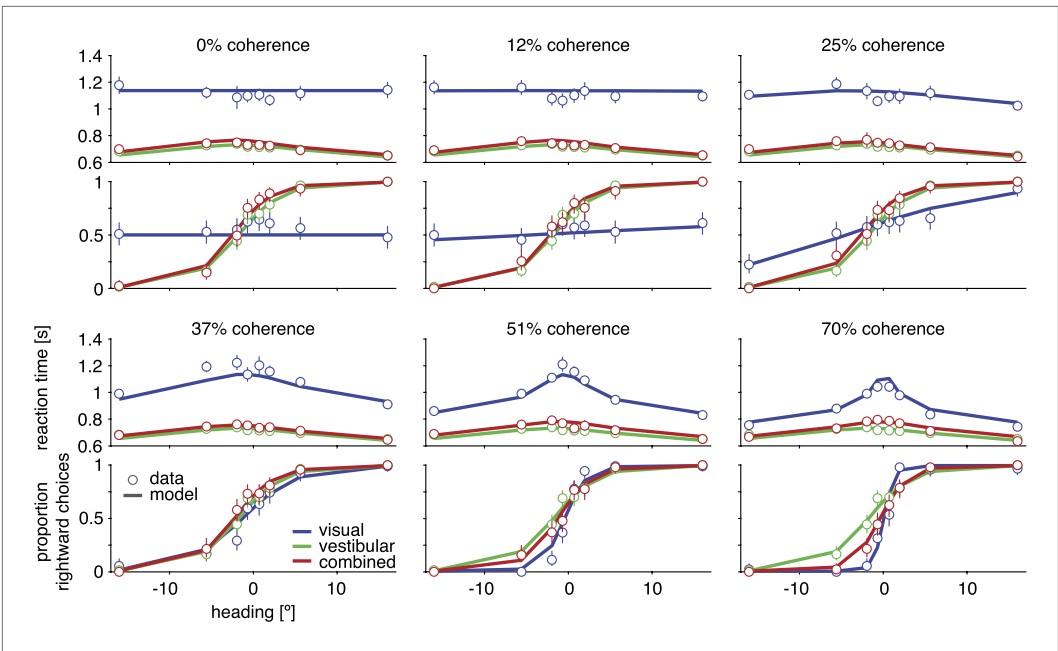

**Figure 3**. Discrimination performance and reaction times for subject D2. Behavioral data (symbols with error bars) and model fits (lines) are shown separately for each motion coherence. Top plot: reaction times as a function of heading; bottom plot: proportion of rightward choices as a function of heading. Mean reaction times are shown for correct trials, with error bars representing two SEM (in some cases smaller than the symbols). Error bars on the proportion rightward choice data are 95% confidence intervals. Although reaction times are only shown for correct trials, the model is fit to data from both correct and incorrect trials. See *Figure 3—figure supplement 1* for behavioral data and model fits for all subjects. *Figure 3—figure supplement 2* shows the fitted model parameters per subject.

The following figure supplements are available for figure 3:

**Figure supplement 1**. Psychometric functions, chronometric functions, and model fits for all subjects.

**Figure supplement 2**. Model parameters for fits of the optimal model and two alternative parameterizations.

vestibular stimulus decreased the discrimination threshold with essentially no loss of speed. A similar overall pattern of results was observed for the other subjects (*Figure 3—figure supplement 1*). These data provide clear evidence that subjects made use of both visual and vestibular information to perform the reaction-time task, but the benefits of cue integration could not be appreciated by considering discrimination thresholds alone.

## Modeling cue combination with a novel diffusion model

To investigate whether subjects accumulate evidence optimally across both time and sensory modalities, we built a model that integrates visual and vestibular cues optimally to perform the heading discrimination task, and we compare predictions of the model to data from our human subjects. The model builds upon the structure of diffusion models (DMs), which have previously been shown to account nicely for the tradeoff between speed and accuracy of decisions (*Ratcliff, 1978*; *Ratcliff and Smith, 2004*; *Palmer et al., 2005*). Additionally, DMs are known to optimally integrate evidence over time (*Laming, 1968*; *Bogacz et al., 2006*), given that the reliability of the evidence is time-invariant (such that, at any point in time from stimulus onset, the stimulus provides the same amount of information about the task variable). However, DMs have neither been used to integrate evidence from several sources, nor to handle evidence whose reliability changes over time, both of which are required for our purposes.

In the context of heading discrimination, a standard DM would operate as follows (*Figure 4A*): consider a diffusing particle with dynamics given by $\dot{x} = k\sin(h) + \eta(t)$, where $h$ is the heading direction,

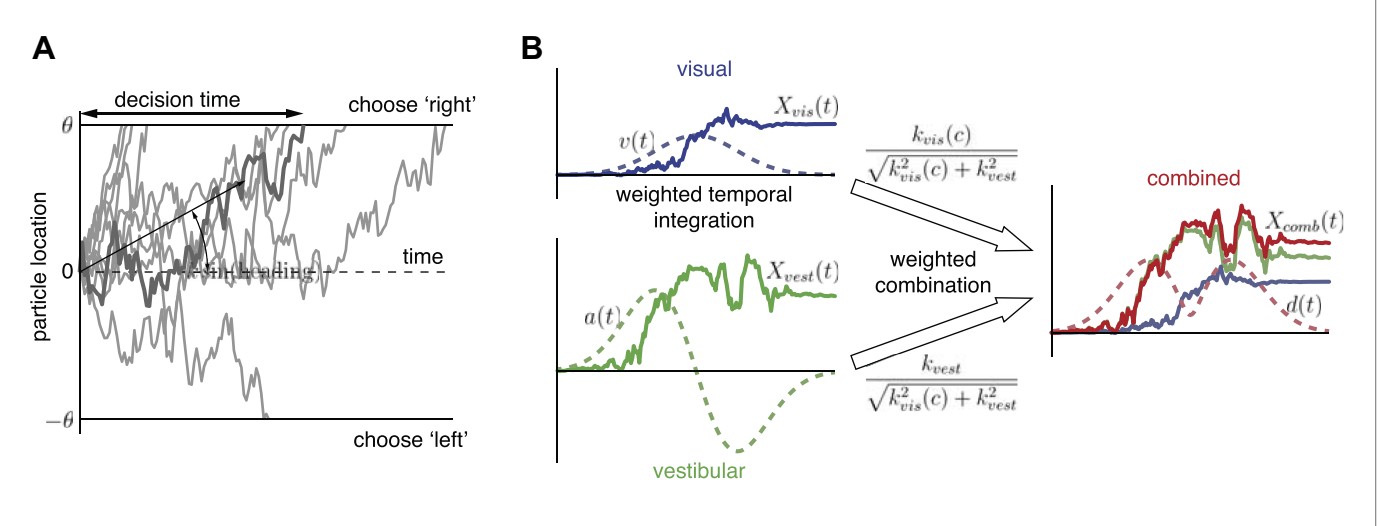

**Figure 4**. Extended diffusion model (DM) for heading discrimination task. (**A**) A drifting particle diffuses until it hits the lower or upper bound, corresponding to choosing 'left' or 'right' respectively. The rate of drift (black arrow) is determined by heading direction. The time at which a bound is hit corresponds to the decision time. 10 particle traces are shown for the same drift rate, corresponding to one incorrect and nine correct decisions. (**B**) Despite time-varying cue sensitivity, optimal temporal integration of evidence in DMs is preserved by weighting the evidence by the momentary measure of its sensitivity. The DM representing the combined condition is formed by an optimal sensitivity-weighted combination of the DMs of the unimodal conditions.

$k$ is a positive constant relating particle drift to heading direction, and $\eta(t)$ is unit variance Gaussian white noise. The particle starts at $x(0) = 0$, drifts with an average slope given by $k\sin(h)$, and diffuses until it hits either the upper bound $\theta$ or the lower bound $-\theta$, corresponding to rightward and leftward choices, respectively. The decision time is determined by when the particle hits a bound. Larger $|h|$'s lead to shorter decision times and more correct decisions because the drift rate is greater. Lower bound levels, $|\theta|$, also lead to shorter decision times but more incorrect decisions. Errors (hitting bound $\theta$ when $h < 0$, or hitting bound $-\theta$ when $h > 0$) can occur due to the stochasticity of particle motion, which is meant to capture the variability of the momentary sensory evidence. The Fisher information in $x(t)$ regarding $h$, a measure of how much information $x(t)$ provides for discriminating heading (**Papoulis, 1991**), is $I_x(\sin(h)) = k^2$ per second, showing that $k$ is a measure of the subject's sensitivity to changes in heading direction. This sensitivity depends on the subject's effectiveness in estimating heading from the cue, which in turn is influenced by the reliability of the cue itself (e.g., coherence).

Now consider both a visual (*vis*) and a vestibular (*vest*) source of evidence regarding $h$, $\dot{x}_{vis} = k_{vis}(c)\sin(h) + \eta_{vis}(t)$ and $\dot{x}_{vest} = k_{vest}\sin(h) + \eta_{vest}(t)$, where $k_{vis}(c)$ indicates that the sensitivity to the cue in the visual modality depends on motion coherence, $c$. Combining these two sources of evidence by a simple sum, $\dot{x}_{vis} + \dot{x}_{vest}$, would amount to adding noise to $\dot{x}_{vest}$ for low coherences ($k_{vis}(c) \approx 0$), which is clearly suboptimal. Rather, it can be shown that the two particle trajectories are combined optimally by weighting their rates of change in proportion to their relative sensitivities (see **Supplementary file 1** for derivation):

$$\dot{x}_{comb} = \sqrt{\frac{k_{vis}^{2}(c)}{k_{vis}^{2}(c) + k_{vest}^{2}}}\,\dot{x}_{vis} + \sqrt{\frac{k_{vest}^{2}}{k_{vis}^{2}(c) + k_{vest}^{2}}}\,\dot{x}_{vest}. \qquad (1)$$

This allows us to model the combined condition by a single new DM, $\dot{x}_{comb} = k_{comb}(c)\sin(h) + \eta_{comb}(t)$, which is optimal because it preserves all information contained in both $x_{vis}$ and $x_{vest}$ (**Figure 4B**; see 'Materials and methods' and **Supplementary file 1** for a formal treatment). The sensitivity (drift rate coefficient) in the combined condition,

$$k_{comb}(c) = \sqrt{k_{vis}^{2}(c) + k_{vest}^{2}}, \qquad (2)$$

is a combination of the sensitivities of the unimodal conditions and is therefore always greater than the largest unimodal sensitivity.

So far we have assumed that the reliability of each cue is time-invariant. However, as the motion velocity changes over time, so does the amount of information about $h$ provided by each cue, and with it the subject's sensitivity to changes in $h$. For the vestibular and visual conditions, motion acceleration $a(t)$ and motion velocity $v(t)$, respectively, are assumed to be the physical quantities that modulate cue sensitivity ('Materials and methods' and 'Discussion'). To account for these dynamics, the DMs are modified to $\dot{x}_{vest} = a(t)k_{vest}\sin(h) + \eta_{vest}(t)$ and $\dot{x}_{vis} = v(t)k_{vis}(c)\sin(h) + \eta_{vis}(t)$. Note that once the drift rate in a DM changes with time, it generally loses its property of integrating evidence optimally over time. For example, at the beginning of each trial when motion velocity is low, $\dot{x}_{vis}$ is dominated by noise and integrating $\dot{x}_{vis}$ is fruitless. Fortunately, weighting the momentary visual evidence, $\dot{x}_{vis}$, by the velocity profile recovers optimality of the DM ('Materials and methods'). This temporal weighting causes the visual evidence to contribute more at high velocities, while the noise is downweighted at low velocities. Similarly, vestibular evidence is weighted by the time course of acceleration. The new, weighted particle trajectories are described by the DMs $\dot{X}_{vis} = v(t)\dot{x}_{vis}$ and $\dot{X}_{vest} = a(t)\dot{x}_{vest}$. The two unimodal DMs are combined as before, resulting in the combined DM given by $\dot{X}_{comb} = d(t)\dot{x}_{comb}$, where the sensitivity profile $d(t)$ is a weighted combination of the unimodal sensitivity profiles,

$$d(t) = \sqrt{\frac{k_{vis}^2(c)}{k_{comb}^2(c)}v^2(t) + \frac{k_{vest}^2}{k_{comb}^2(c)}a^2(t)}. \tag{3}$$

(*Figure 4B*; see *Supplementary file 1* for derivation). These modifications to the standard DM are sufficient to integrate evidence optimally across time and sensory modalities, even as the sensitivity to the evidence changes over time.

The model assumes that subjects know their cue sensitivities, $k_{vis}(c)$ and $k_{vest}$, as well as the temporal sensitivity profiles, $a(t)$ and $v(t)$, of each stimulus. In this respect, our model provides an upper bound on performance, since subjects may not have perfect knowledge of these variables, especially since stimulus modalities and visual motion coherence values are randomized across trials ('Discussion').

## Quantitative assessment of cue combination performance

We tested whether subjects combined evidence optimally across both time and cues by evaluating how well the model outlined above could explain the observed behavior. The bounds, $\theta$, of the modified DM, and the sensitivity parameters ($k_{vis}$, $k_{vest}$ and $k_{comb}$), were allowed to vary between the visual, vestibular, and combined conditions. Varying the bound was essential to capture the deviation of the discrimination threshold in the combined condition from that predicted by traditional cue combination models (*Figure 2*). Indeed, this discrimination threshold is inversely proportional to bound and sensitivity (see *Supplementary file 1*). Since the sensitivity in the bimodal condition is not a free parameter (it is determined by *Equation 2*), the height of the bound is the only parameter that could modulate the discrimination thresholds.

The noise terms $\eta_{vis}$ and $\eta_{vest}$ play crucial roles in the model, as they relate to the reliability of the momentary sensory evidence. To specify the manner in which such noise may depend on motion coherence, we relied on fundamental assumptions about how optic flow stimuli are represented by the brain. We assumed that heading is represented by a neural population code in which neurons have heading tuning curves that, within the range of heading tested in this experiment (±16°, *Figure 5A*), differ in their heading preferences but have similar shapes. This is broadly consistent with data from area MSTd (*Fetsch et al., 2011*), but the exact location of such a code is not important for our argument. For low coherence, motion energy in the stimulus is almost uniform for all heading directions, such that all neurons in the population fire at approximately the same rate (*Figure 5A*, dark blue curve). For high coherence, population neural activity is strongly peaked around the actual heading direction (*Figure 5A*, cyan curve) (*Morgan et al., 2008*; *Fetsch et al., 2011*).

Based on this representation, and assuming that the response variability of the neurons belongs to the exponential family with linear sufficient statistics (*Ma et al., 2006*) (an assumption consistent with in vivo data [*Graf et al., 2011*]), heading discrimination can be performed optimally by a weighted sum of the activity of all neurons, with weights monotonically related to the preferred heading of each neuron. For a straight forward heading, $h = 0$, this sum should be 0, and for $h > 0$ (or $h < 0$) it should be positive (or negative), thus sharing the basic properties of the momentary evidence, $\dot{x}$, in our DM. This allowed us to deduce the mean and variance of the momentary evidence driving $\dot{x}$, based on what

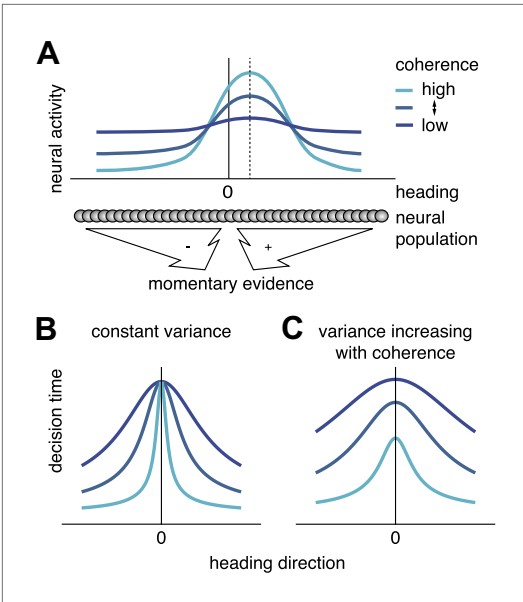

**Figure 5**. Scaling of momentary evidence statistics of the diffusion model (DM) with coherence. (**A**) Assumed neural population activity giving rise to the DM mean and variance of the momentary evidence, and their dependence on coherence. Each curve represents the activity of a population of neurons with a range of heading preferences, in response to optic flow with a particular coherence and a heading indicated by the dashed vertical line. (**B**) Expected pattern of reaction times if variance is independent of coherence. If neither the DM bound nor the DM variance depend on coherence, the DM predicts the same decision time for all small headings, regardless of coherence. This is due to the DM drift rate, $k_{vis}(c)sin(h)$ being close to 0 for small headings, $h \approx 0$, independent of the DM sensitivity $k_{vis}(c)$. (**C**) Expected pattern of reaction times when variance scales with coherence. If both DM sensitivity and DM variance scale with coherence while the bound remains constant, the DM predicts different decision times across coherences, even for small headings. Greater coherence causes an increase in variance, which in turn causes the bound to be reached more quickly for higher coherences, even if the heading, and thus the drift rate, is small.

we know about the neural responses. First, the sensitivity, $k_{vis}(c)$, which determines how optic flow modulates the mean drift rate of $\dot{x}$, scales in proportion with the 'peakedness' of the neural activity, which in turn is proportional to coherence. We assumed a functional form of $k_{vis}(c)$ given by $a_{vis}c^{\gamma_{vis}}$, where $a_{vis}$ and $\gamma_{vis}$ are positive parameters. Second, the variance of $\dot{x}$ is assumed to be the sum of the variances of the neural responses. Since experimental data suggest that the variance of these responses is proportional to their firing rate (**Tolhurst et al., 1983**), the sum of the variances is proportional to the area underneath the population activity profile (**Figure 5A**). Based on the experimental data of Britten et al. (**Heuer and Britten, 2007**), this area was assumed to scale roughly linearly with coherence, such that the variance of $\dot{x}$ is proportional to $1 + b_{vis}c^{\gamma_{vis}}$ with free parameters $b_{vis}$ and $\gamma_{vis}$, the latter of which captures possible deviations from linearity. We further assumed the DM bound to be independent of coherence, and given by $\theta_{\sigma,vis}$. Thus, the effect of motion coherence on the momentary evidence in the DM was modeled by four parameters: $a_{vis}$, $\gamma_{vis}$, $b_{vis}$, and $\theta_{\sigma,vis}$.

The above scaling of the diffusion variance by coherence, which is a consequence of the neural code for heading, makes an interesting prediction: reaction times for headings near straight ahead should be inversely proportional to coherence in the visual condition, even though the mean drift rate, $k_{vis}(c)sin(h)$, is very close to 0. This is indeed what we observed: subjects tended to decide faster for higher coherences even when $h \approx 0$ (**Figure 3**, **Figure 3—figure supplement 1**). This aspect of the data can only be captured by the model if the DM variance is allowed to change with coherence (**Figure 5B,C**).

To summarize, in the combined condition, the diffusion variance was assumed to be proportional to $1 + b_{comb}c^{\gamma_{comb}}$, while the bound was fixed at $\theta_{\sigma,comb}$. By contrast, the diffusion rate (sensitivity) cannot be modeled freely but rather needs to obey $k_{comb}(c) = \sqrt{k_{vis}^2(c) + k_{vest}^2}$ in order to ensure optimal cue combination. The sensitivity $k_{vest}$ and bound $\theta_{\sigma,vest}$ in the vestibular condition do not depend on motion coherence and were thus model parameters that were fitted directly.

Observed reaction times were assumed to be composed of the decision time and some non-decision time. The decision time is the time from the start of integrating evidence until a decision is made, as predicted by the diffusion model. The non-decision time includes the motor latency and the time from stimulus onset to the start of integrating evidence. As the latter can vary between different modalities, we allowed it to differ between visual, vestibular, and combined conditions, but not for different coherences, thus introducing the model parameters $t_{nd,vis}$, $t_{nd,vest}$, and $t_{nd,comb}$. Although the fitted non-decision times were similar across stimulus conditions for most subjects (**Figure 3—figure supplement 2**), a model assuming a single non-decision time resulted in a small but significant decrease in fit quality (**Figure 7—figure supplement 2A**). Overall, 12 parameters were used to model

cue sensitivities, bounds, variances, and non-decision times in all conditions, and these 12 parameters were used to fit 312 data points for subjects that were tested with 6 coherences (168 data points for the three-coherence version). An additional 14 parameters (8 parameters for the three-coherence version; one bias parameter per coherence/condition, one lapse parameter across all condition) controlled for biases in the motion direction percept and for lapses of attention that were assumed to lead to random choices ('Materials and methods'). Although these additional parameters were necessary to achieve good model fits (*Figure 7—figure supplement 2A*), it is critical to note that they could not account for differences in heading thresholds or reaction times across stimulus conditions. As such, the additional parameters play no role in determining whether subjects perform optimal multisensory integration. Alternative parameterizations of how drift rates and bounds depend on motion coherence yielded qualitatively similar results, but caused the model fits to worsen decisively (*Supplementary file 1*; *Figure 7—figure supplement 2A*).

Critically, our model predicts that the unimodal sensitivities $k_{vis}(c)$ and $k_{vest}$ relate to the combined value by $k_{comb}^{predicted}(c) = \sqrt{k_{vis}(c)^2 + k_{vest}^2}$, if subjects accumulate evidence optimally across cues. To test this prediction, we fitted separately the unimodal and combined sensitivities, $k_{vis}(c)$, $k_{vest}$ and $k_{comb}$ to the complete data set from each individual subject using maximum likelihood optimization ('Materials and methods'), and then compared the fitted values of $k_{comb}$ to the predicted values, $k_{comb}^{predicted}(c)$. Predicted and observed sensitivities for the combined condition are virtually identical (*Figure 6*), providing strong support for near-optimal cue combination across both time and cues. Remarkably, for low coherences at which optic flow provides no useful heading information, the sensitivity in the combined condition was not significantly different from that of the vestibular condition (*Figure 6*). Thus, subjects were able to completely suppress noisy visual information and rely solely on vestibular input, as predicted by the model.

Having established that cue sensitivities combine according to *Equation 2*, the model was then fit to data from each individual subject under the assumption of optimal cue combination. Model fits are shown as solid curves for example subject D2 (*Figure 3*), as well as for all other subjects (*Figure 3—figure supplement 1*). Sensitivity parameters, bounds, and non-decision times resulting from the fits are also shown for each subject, condition, and coherence (*Figure 3—figure supplement 2*). For 8 of 10 datasets, the model explains more than 95% of the variance in the data (adjusted $R^2 > 0.95$), providing additional evidence for near-optimal cue combination across both time and cues (*Figure 7A*). The subjects associated with these datasets show a clear decrease in reaction times with larger $|h|$, and this effect is more pronounced in the visual condition than in the vestibular and combined conditions (*Figure 3*, *Figure 3—figure supplement 1*). The remaining two subjects (C and F) feature qualitatively different behavior and lower $R^2$ values of approximately 0.80 and 0.90, respectively (*Figure 3—figure supplement 1*). These subjects showed little decline in reaction times with larger values of $|h|$, and their mean reaction times were more similar across the visual, vestibular and combined conditions.

Critically, the model nicely captures the observation that the psychophysical threshold in the combined condition is typically greater than that for the visual condition, despite near-optimal combination of momentary evidence from the visual and vestibular modalities (e.g., *Figure 3*, 70% coherence, *Figure 2—figure supplement 1*,

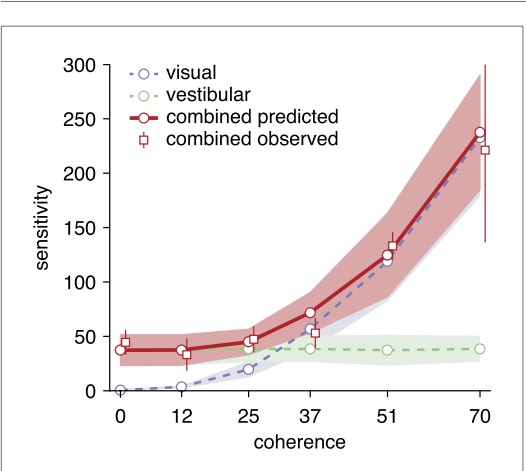

**Figure 6**. Predicted and observed sensitivity in the combined condition. The sensitivity parameter measures how sensitive subjects are to a change of heading. The solid red line shows predicted sensitivity for the combined condition, as computed from the sensitivities of the unimodal conditions (dashed lines). The combined sensitivity measured by fitting the model to each coherence separately (red squares) does not differ significantly from the optimal prediction, providing strong support to the hypothesis that subjects accumulate evidence near-optimally across time and cues. Data are averaged across datasets (except 0%, 12%, 51% coherence: only datasets B2, D2, F2), with shaded areas and error bars showing the 95% CIs. DOI: 10.7554/eLife.03005.011

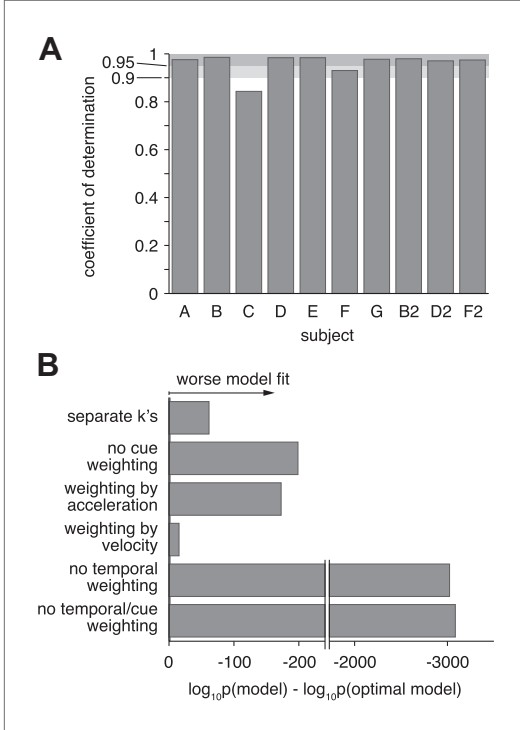

**Figure 7**. Model goodness-of-fit and comparison to alternative models. (**A**) Coefficient of determination (adjusted $R^2$) of the model fit for each of the ten datasets. (**B**) Bayes factor of alternative models compared to the optimal model. The abscissa shows the base-10 logarithm of the Bayes factor of the alterative models vs the optimal model (negative values mean that the optimal model out-performs the alternative model). The gray vertical line close to the origin (at a value of −2 on the abscissa) marks the point at which the optimal model is 100 times more likely than each alternative, at which point the difference is considered 'decisive' (**Jeffreys, 1998**). Only the 'separate k's' model has more parameters than the optimal model, but the Bayes factor indicates that the slight increase in goodness-of-fit does not justify the increased degrees of freedom. The 'no cue weighting' model assumes that visual and vestibular cues are weighted equally, independent of their sensitivities. The 'weighting by acceleration' and 'weighting by velocity' models assume that the momentary evidence of both cues is weighted by the acceleration and velocity profile of the stimulus, respectively. The 'no temporal weighting' model assumes that the evidence is not weighted over time according to its sensitivity. The 'no cue/temporal weighting' model lacks both weighting of cues by sensitivity and weighting by temporal profile. All of the tested alternative models explain the data decisively worse than the optimal model. **Figure 7—figure supplement 1** shows how individual subjects contribute to this model comparison, and the results of a more conservative Bayesian random-effects model comparison that supports same conclusion. **Figure 7—figure supplement 2**

*Figure 7. Continued on next page*

*Figure 3—figure supplement 1*). Thus, the model fits confirm quantitatively that apparent sub-optimality in psychophysical thresholds can arise even if subjects combine all cues in a statistically optimal manner, emphasizing the need for a computational framework that incorporates both decision accuracy and speed.

## Alternative models

To further assess and validate the critical design features of our modified DM, we evaluated six alternative (mostly sub-optimal) versions of the model to see if these variants are able to explain the data equally well. We compared these variants to the optimal model using Bayesian model comparison, which trades off fit quality with model complexity to determine whether additional parameters significantly improve the fit (**Goodman, 1999**).

With regard to optimality of cue integration across modalities, we examined two model variants. The first variant (also used to generate **Figure 6**) eliminates the relationship, $k_{comb}(c) = \sqrt{k_{vis}^2(c) + k_{vest}^2}$ (**Equation 2**), between the sensitivity parameters in the combined and single-cue conditions. Instead, this variant allows independent sensitivity parameters for the combined condition at each coherence, thus introducing one additional parameter per coherence. Since this variant is strictly more general than the optimal model, it must fit the data at least as well. However, if the subjects' behavior is near optimal, the additional degrees of freedom in this variant should not improve the fit enough to justify the addition of these parameters. This is indeed what we found by Bayesian model comparison (**Figure 7B**, 'separate k's'), which shows the optimal model to be ~$10^{70}$ times more likely than the variant with independent values of $k_{comb}(c)$. This is well above the threshold value that is considered to provide 'decisive' evidence in favor of the optimal model (we use Fisher's definition of decisive [**Jeffreys, 1998**] according to which a model is said to be decisively better if it is >100 times more likely to have generated the data). The second model variant had the same number of parameters as the optimal model, but assumed that the cues are always weighted equally. Evidence in the combined condition was given by the simple average, $\dot{X}_{comb} = \frac{1}{2}\left(\dot{X}_{vis} + \dot{X}_{vest}\right)$, ignoring cue sensitivities. The resulting fits (**Figure 7B**, 'no cue weighting') are also decisively worse than those of the optimal model. Together, these model variants strongly support the hypothesis that subjects weight cues according to their relative

*Figure 7. Continued*

compares the proposed model to ones with alternative parameterizations.

The following figure supplements are available for figure 7:

**Figure supplement 1**. Model comparison per subject, and random-effects model comparison.

**Figure supplement 2**. Model comparison for models with alternative parameterization.

sensitivities, as given by *Equation 2*. These effects were largely consistent across individual subjects (*Figure 7—figure supplement 1A*).

To test the other key assumption of our model—that subjects temporally weight incoming evidence according to the profile of stimulus information—we tested three model variants that modified how temporal weighting was performed without changing the number of parameters in the model. If we assumed that the temporal weighting of both modalities followed the acceleration profile of the stimulus while leaving the model otherwise unchanged, the model fit worsened decisively (*Figure 7B*, 'weighting by acceleration'). Assuming that the weighting of both modalities followed the velocity profile of the stimulus also decisively reduced fit quality (*Figure 7B*, 'weighting by velocity'), although this effect was not consistent across subjects (*Figure 7—figure supplement 1A*). If we completely removed temporal weighting of cues from the model, fits were dramatically worse than the optimal model (*Figure 7B*, 'no temporal weighting'). Finally, for completeness, we also tested a model variant that neither performs temporal weighting of cues nor considers the relative sensitivity to the cues. Again, this model variant fit the data decisively worse than the optimal model (*Figure 7B*, 'no cue/temporal weighting'). Thus, subjects seem to be able to take into account their sensitivity to the evidence across time as well as across cues. All of these model comparisons received further support from a more conservative random-effects Bayesian model comparison, shown in *Figure 7—figure supplement 1B,C*.

Finally, we also considered if a parallel race model could account for our data. The parallel race model (*Raab, 1962*; *Miller, 1982*; *Townsend and Wenger, 2004*; *Otto and Mamassian, 2012*) postulates that the decision in the combined condition emerges from the faster of two independent races toward a bound, one for each sensory modality. Because it does not combine information across modalities, the parallel race model predicts that decisions in the combined condition are caused by the faster modality. Consequently, choices in the combined condition are unlikely to be more correct (on average) than those of the faster unimodal condition. For all but one subject, the vestibular modality is substantially faster, even when compared to the visual modality at high coherence and controlling for the effect of heading direction (2-way ANOVA, $p<0.0001$ for all subjects except C). Critically, all of these subjects feature significantly lower psychophysical thresholds in the combined condition than in the vestibular condition ($p<0.039$ for all subjects except subject C, $p=0.210$, *Supplementary file 2A*). Furthermore, we performed standard tests (Miller's bound and Grice's bound) that compare the observed distribution of reaction times with that predicted by the parallel race model (*Miller, 1982*; *Grice et al., 1984*). These tests revealed that all but two subjects made significantly slower decisions than predicted by the parallel race model for most coherence/heading combinations ($p<0.05$ for all subjects except subjects F and B2; *Supplementary file 2B*), and no subject was faster than predicted ($p>0.05$, all subjects; *Supplementary file 2B*). Based on these observations, we can reject the parallel race model as a viable hypothesis to explain the observed behavior.

## Discussion

We have shown that, when subjects are allowed to choose how long to accumulate evidence in a cue integration task, their behavior no longer follows the standard predictions of optimal cue integration theory that normally apply when stimulus presentation time is controlled by the experimenter. Particularly, they feature worse discrimination performance (higher psychophysical thresholds) in the combined condition than would be predicted from the unimodal conditions—in some cases even worse than the better of the two unimodal conditions. This occurs because subjects tend to decide more quickly in the combined condition than in the more sensitive unimodal condition and thus have less time to accumulate evidence. This indicates that a more general definition of optimal cue integration must incorporate reaction times. Indeed, subjects' behavior could be reproduced by an extended diffusion model that takes into account both speed and accuracy, thus suggesting that subjects accumulate evidence across both time and cues in a statistically near-optimal manner (i.e., with minimal information loss) despite their reduced discrimination performance in the combined condition.

Previous work on optimal cue integration (e.g., *Ernst and Banks, 2002*; *Battaglia et al., 2003*; *Knill and Saunders, 2003*; *Fetsch et al., 2009*) was based on experiments that employed fixed-duration stimuli and was thus able to ignore how subjects accumulate evidence over time. Moreover, previous work relied on the implicit assumption that subjects make use of all evidence throughout the duration of the stimulus. However, this assumption need not be true and has been shown to be violated even for short presentation durations (*Mazurek et al., 2003*; *Kiani et al., 2008*). Therefore, apparent sub-optimality in some previous studies of cue integration or in some individual subjects (*Battaglia et al., 2003*; *Fetsch et al., 2009*) might be attributable to either truly sub-optimal cue combination, to subjects halting evidence accumulation before the end of the stimulus presentation period, or to the difficulty in estimating stimulus processing time (*Stanford et al., 2010*). Unfortunately, these potential causes cannot be distinguished using a fixed-duration task. Allowing subjects to register their decisions at any time during the trial alleviates this potential confound.

We model subjects' decision times by assuming an accumulation-to-bound process. In the multisensory context, this raises the question of whether evidence accumulation is bounded for each modality separately, as assumed by the parallel race model, or whether evidence is combined across modalities before being accumulated toward a single bound, as in co-activation models and our modified diffusion model. Based on our behavioral data, we can rule out parallel race models, as they cannot explain lower psychophysical thresholds (better sensitivity) in the combined condition relative to the faster vestibular condition. Further evidence against such models is provided by neurophysiological studies which demonstrate that visual and vestibular cues to heading converge in various cortical areas, including areas MSTd (*Gu et al., 2006*), VIP (*Schlack et al., 2005*; *Chen et al., 2011b*), and VPS (*Chen et al., 2011a*). Activity in area MSTd can account for sensitivity-based cue weighting in a fixed-duration task (*Fetsch et al., 2011*), and MSTd activity is causally related to multi-modal heading judgments (*Britten and van Wezel, 1998*, *2002*; *Gu et al., 2012*). These physiological studies strongly suggest that visual and vestibular signals are integrated in sensory representations prior to decision-making, inconsistent with parallel race models.

Our model makes the assumption that sensory signals are integrated prior to decision-making and is in this sense similar to co-activation models that have been used previously to model reaction times in multimodal settings (*Miller, 1982*; *Corneil et al., 2002*; *Townsend and Wenger, 2004*). However, it differs from these models in important aspects. First, co-activation models have been introduced to explain reaction times that are faster than those predicted by parallel race models (*Raab, 1962*; *Miller, 1982*). Our subjects, in contrast, feature reaction times that are slower than those of parallel race models in almost all conditions (*Supplementary file 2B*). We capture this effect by an elevated effective bound in the combined condition as compared to the faster vestibular condition, such that cue combination remains optimal despite longer reaction times. Second, co-activation models usually combine inputs from the different modalities by a simple sum (e.g., *Townsend and Wenger, 2004*). This entails adding noise to the combined signal if the sensitivity to one of the modalities is low, which is detrimental to discrimination performance. In contrast, we show that different cues need to be weighted according to their sensitivities to achieve statistically optimally integration of multisensory evidence at each moment in time (*Equation 2*).

Another alternative to co-activation models are serial race models, which posit that the race corresponding to one cue needs to be completed before the other one starts (e.g., *Townsend and Wenger, 2004*). These models can be ruled out by observing that they predict reaction times in the combined condition to be longer than those in the slower of the two unimodal conditions. This is clearly violated by the subjects' behavior.

Optimal accumulation of evidence over time requires the momentary evidence to be weighted according to its associated sensitivity. For the vestibular modality, we assume that the temporal profile of sensitivity to the evidence follows acceleration. This may appear to conflict with data from multimodal areas MSTd, VIP, and VPS, where neural activity in response to self-motion reflects a mixture of velocity and acceleration components (*Fetsch et al., 2010*; *Chen et al., 2011a*). Note, however, that the vestibular stimulus is initially encoded by otolith afferents in terms of acceleration (*Fernandez and Goldberg, 1976*). Thus, any neural representation of vestibular stimuli in terms of velocity requires a temporal integration of the acceleration signal, and this integration introduces temporal correlations into the signal. As a consequence, a neural response that is maximal at the time of peak stimulus velocity does not imply a simultaneous peak in the information coded about heading direction. Rather, information still follows the time course of its original encoding, which is in terms of acceleration.

In contrast, the time course of the sensitivity to the visual stimulus is less clear. For our model we have intuitively assumed it to follow the velocity profile of the stimulus, as information per unit time about heading certainly increases with the velocity of the optic flow field, even when there is no acceleration. This assumption is supported by a decisively worse model fit if we set the weighting of the visual momentary evidence to follow the acceleration profile (*Figure 7B*, 'weighted by acceleration'). Nonetheless, we cannot completely exclude any contribution of acceleration components to visual information (*Lisberger and Movshon, 1999*; *Price et al., 2005*). In any case, our model fits make clear that temporal weighting of vestibular and visual inputs is necessary to predict behavior when stimuli are time-varying.

The extended DM model described here makes the strong assumption that cue sensitivities are known before combining information from the two modalities, as these sensitivities need to be known in order to weight the cues appropriately. As only the sensitivity to the visual stimulus changes across trials in our experiment, it is possible that subjects can estimate their sensitivity (as influenced by coherence) during the initial low-velocity stimulus period (*Figure 1C*) in which heading information is minimal but motion coherence is salient. Thus, for our task, it is reasonable to assume that subjects can estimate their sensitivity to cues. We have recently begun to consider how sensitivity estimation and cue integration can be implemented neurally. The neural model (*Onken et al., 2012*. Near optimal multisensory integration with nonlinear probabilistic population codes using divisive normalization. The Society for Neuroscience annual meeting 2012) estimates the sensitivity to the visual input from motion sensitive neurons and uses this estimate to perform near-optimal multisensory integration with generalized probabilistic population codes (*Ma et al., 2006*; *Beck et al., 2008*) using divisive normalization. We intend to extend this model to the integration of evidence over time to predict neural responses (e.g., in area LIP) that should roughly track the temporal evolution of the decision variable ($x_{comb}(t)$, 'Materials and methods') in the DM model. This will make predictions for activity in decision-making areas that can be tested in future experiments.

In closing, our findings establish that conventional definitions of optimality do not apply to cue integration tasks in which subjects' decision times are unconstrained. We establish how sensory evidence should be weighted across modalities and time to achieve optimal performance in reaction-time tasks, and we show that human behavior is broadly consistent with these predictions but not with alternative models. These findings, and the extended diffusion model that we have developed, provide the foundation for building a general understanding of perceptual decision-making under more natural conditions in which multiple cues vary dynamically over time and subjects make rapid decisions when they have acquired sufficient evidence.

## Materials and methods

### Subjects and apparatus

Seven subjects (3 males) aged 23–38 years with normal or corrected-to-normal vision and no history of vestibular deficits participated in the experiments. All subjects but one were informed of the purposes of the study. Informed consent was obtained from all participants and all procedures were reviewed and approved by the Washington University Office of Human Research Protections (OHRP), Institutional Review Board (IRB; IRB ID# 201109183). Consent to publish was not obtained in writing, as it was not required by the IRB, but all subjects were recruited for this purpose and approved verbally. Of these subjects, three (subjects B, D, F; 1 male) participated in a follow-up experiment roughly 2 years after the initial data collection, with six coherence levels instead of the original three. The six-coherence version of their data is referred to as B2, D2, and F2. Procedures for the follow-up experiment were approved by the Institutional Review Board for Human Subject Research for Baylor College of Medicine and Affiliated Hospitals (BCM IRB, ID# H-29411) and informed consent and consent to publish was given again by all three subjects.

The apparatus, stimuli, and task design have been described in detail previously (*Fetsch et al., 2009*; *Gu et al., 2010*), and are briefly summarized here. Subjects were seated comfortably in a padded racing seat that was firmly attached to a 6-degree-of-freedom motion platform (MOOG, Inc). A 3-chip DLP projector (Galaxy 6; Barco, Kortrijk, Belgium) was mounted on the motion platform behind the subject and front-projected images onto a large (149 × 127 cm) projection screen via a mirror mounted above the subject's head. The viewing distance to the projection screen was ~70 cm, thus allowing for a field of view of ~94° × 84°. Subjects were secured to the seat using a 5-point racing

harness, and a custom-fitted plastic mask immobilized the head against a cushioned head mount. Seated subjects were enclosed in a black aluminum superstructure, such that only the display screen was visible in the darkened room. To render stimuli stereoscopically, subjects wore active stereo shutter glasses (CrystalEyes 3; RealD, Beverly Hills, CA) which restricted the field of view to ~90° × 70°. Subjects were instructed to look at a centrally-located, head-fixed target throughout each trial. Sounds from the motion platform were masked by playing white noise through headphones. Behavioral task sequences and data acquisition were controlled by Matlab and responses were collected using a button box.

Visual stimuli were generated by an OpenGL accelerator board (nVidia Quadro FX1400), and were plotted with sub-pixel accuracy using hardware anti-aliasing. In the visual and combined conditions, visual stimuli depicted self-translation through a 3D cloud of stars distributed uniformly within a virtual space 130 cm wide, 150 cm tall, and 75 cm deep. Star density was $0.01/cm^3$, with each star being a 0.5 cm × 0.5 cm triangle. Motion coherence was manipulated by randomizing the three-dimensional location of a percentage of stars on each display update while the remaining stars moved according to the specified heading. The probability of a single star following the trajectory associated with a particular heading for N video updates is therefore $(c/100)^N$, where c denotes motion coherence (ranging from 0–100%). At the largest coherence used here (70%), there is only a 3% probability that a particular star would follow the same trajectory for 10 display updates (0.17 s). Thus, it was practically not possible for subjects to track the trajectories of individual stars. This manipulation degraded optic flow as a heading cue and was used to manipulate visual cue reliability in the visual and combined conditions. 'Zero' coherence stimuli had c set to 0.1, which was practically indistinguishable from c = 0, but allowed us to maintain a precise definition of the correctness of the subject's choice.

## Behavioral task

In all stimulus conditions, the task was a single-interval, two-alternative forced choice (2AFC) heading discrimination task. In each trial, human subjects were presented with a translational motion stimulus in the horizontal plane (Gaussian velocity profile; peak velocity, 0.403 m/s; peak acceleration, $0.822 m/s^2$; total displacement, 0.3 m; maximum duration, 2 s). Heading was varied in small steps around straight ahead (±0.686°, ±1.96°, ±5.6°, ±16°) and subjects were instructed to report (by a button press) their perceived heading (leftward or rightward relative to an internal standard of straight ahead) as quickly and accurately as possible. In the visual and combined conditions, cue reliability was varied across trials by randomly choosing the motion coherence of the visual stimulus from among either a group of three values (25%, 37%, and 70%, subjects A–G) or a group of six values (0%, 12%, 25%, 37%, 51%, and 70%, subjects B2, D2, F2). A coherence of 25% means that 25% of the dots move in a direction consistent with the subject's heading, whereas the remaining 75% of the dots are relocated randomly within the dot cloud. In the combined condition, visual and vestibular stimuli always specified the same heading (there was no cue conflict).

During the main phase of data collection, subjects were not informed about the correctness of their choices (no feedback). In the vestibular and combined conditions, platform motion was halted smoothly but rapidly immediately following registration of the decision, and the platform then returned to its original starting point. In the visual condition, the optic flow stimulus disappeared from the screen when a decision was made. In all conditions, 2.5 s after the decision, a sound informed the subjects that they could initiate the next trial by pushing a third button. Once a trial was initiated, the stimulus onset occurred following a randomized delay period (truncated exponential; mean, 987 ms). Prior to data collection, subjects were introduced to the task for 1–2 week 'training' sessions, in which they were informed about the correctness of their choices by either a low-frequency (incorrect) or a high-frequency (correct) sound. The training period was terminated once their behavior stabilized across consecutive training sessions. During training, subjects were able to adjust their speed-accuracy trade-off based on feedback. During subsequent data collection, we did not observe any clear changes in the speed-accuracy trade-off exhibited by subjects.

## Data analysis

Analyses and statistical tests were performed using MATLAB R2013a (The Mathworks, MA, USA).

For each subject, discrimination thresholds were determined separately for each combination of stimulus modality (visual-only, vestibular-only, combined) and coherence (25%, 37%, and 70% for

subjects A–G; 0%, 12%, 25%, 37%, 51%, and 70% for subjects B2, D2, F2) by plotting the proportion of rightward choices as a function of heading direction (*Figure 2A*). The psychophysical discrimination threshold was taken as the standard deviation of a cumulative Gaussian function, fitted by maximum likelihood methods. We assumed a common lapse rate (proportion of random choices) across all stimulus conditions, but allowed for a separate bias parameter (horizontal shift of the psychometric function) for each modality/coherence. Confidence intervals for threshold estimates were obtained by taking 5000 parametric bootstrap samples (*Wichmann and Hill, 2001*). These samples also form the basis for statistical comparisons of discrimination thresholds: two thresholds were compared by computing the difference between their associated samples, leading to 5000 threshold difference samples. Subsequently, we determined the fraction of differences that were below or above zero, depending on the directionality of interest. This fraction determined the raw significance level for accepting the null hypothesis (no difference). The reported significance levels are Bonferroni-corrected for multiple comparisons. All comparisons were one-tailed. Following traditional cue combination analyses (*Clark and Yuille, 1990*), the optimal threshold $\sigma_{pred,c}$ in the combined condition for coherence $c$ was predicted from the visual threshold $\sigma_{vis,c}$ and the vestibular threshold $\sigma_{vest}$ by $\sigma^2_{pred,c} = \sigma^2_{vis,c}\sigma^2_{vest}/(\sigma^2_{vis,c} + \sigma^2_{vest})$. Confidence intervals and statistical tests were again based on applying this formula to individual bootstrap samples of the unimodal threshold estimates. *Supplementary file 2A* reports the p-values for all subjects and all comparisons.

For each dataset, we evaluated the absolute goodness-of-fit of the optimal model (*Figure 7A*) by finding the set of model parameters $\varphi$ that maximized the likelihood of the observed choices and reaction times, and then computing the average coefficient of determination, $R^2(D\varphi) = \frac{1}{2}\left(R^2_{psych}(\varphi) + R^2_{chron}(\varphi)\right)$. Here, $R^2_{psych}(\varphi)$ and $R^2_{chron}(\varphi)$ denote the adjusted $R^2$ values for the psychometric and chronometric functions, respectively, across all modalities/coherences. The value of $R^2_{psych}$ for the psychometric function was based on the probability of making a correct choice across all heading angles, coherences, and conditions, weighted by the number of observations, and adjusted for the number of model parameters. The same procedure, based on the mean reaction times, was used to find $R^2_{chron}$, but we additionally distinguished between mean reaction times for correct and incorrect choices, and fitted both weighted by their corresponding number of observations (see SI for expressions for $R^2_{psych}(\varphi)$ and $R^2_{chron}(\varphi)$).

We compared different variants of the full model (*Figure 7B*) by Bayesian model comparison based on Bayes factors, which were computed as follows. First, we found for each model $\mathcal{M}$ and subject $s$ the set of parameters $\varphi$ that maximized the likelihood, $\varphi^*_{s,\mathcal{M}} = \arg\max_\varphi p(\text{data of subj } s\,|\,\varphi,\mathcal{M})$. Second, we approximated the Bayesian model evidence, measuring the model posterior probability while marginalizing over the parameters, up to a constant by the Bayesian information criterion, $\ln p(\mathcal{M}\,|\,s) \approx -\frac{1}{2}\text{BIC}(s,\mathcal{M})$ with $\text{BIC}(s,M) = -2\ln p(s\,|\,\varphi^*_{s,\mathcal{M}},\mathcal{M}) + k_\mathcal{M}\ln N_s$. Here, $k_\mathcal{M}$ is the number of parameters of model $\mathcal{M}$, and $N_s$ is the number of trials for dataset $s$, respectively. Based on this, we computed the Bayes factor of model $\mathcal{M}$ vs the optimal model $\mathcal{M}_{opt}$ by pooling the model evidence over datasets, resulting in $\sum_s \left(\ln p(\mathcal{M}\,|\,s) - \ln p(\mathcal{M}_{opt}\,|\,s)\right)$. These values, converted to a base-10 logarithm, are shown in *Figure 7B*. In this case, a negative $\log_{10}$-difference of 2 implies that the optimal model is 100 times more likely given the data than the alternative model, a difference that is considered decisive in favor of the optimal model (*Jeffreys, 1998*).

To determine the faster stimulus modality for each subject, we compared reaction times for the vestibular condition with those for the visual condition at 70% coherence. We tested the difference in the logarithm of these reaction times by a 2-way ANOVA with stimulus modality and heading direction as the two factors, and we report the main effect of stimulus modality on reaction times. Although we performed a log-transform of the reaction times to ensure their normality, a Jarque–Bera test revealed that normality did not hold for some heading directions. Thus, we additionally performed a Friedman test on subsampled data (to have the same number of trials per modality/heading) which supported the ANOVA result at the same significance level. In the main text, we only report the main effect of stimulus modality on reaction time from the 2-way ANOVA. Detailed results of the 2-way ANOVA, the Jarque–Bera test, and the Friedman test are reported for each subject in *Supplementary file 2C*.

## The extended diffusion model

Here we outline the critical extensions to the diffusion model. Detailed derivations and properties of the model are described in the *Supplementary file 1*.

Discretizing time into small steps of size $\Delta$ allows us to describe the particle trajectory $x(t)$ in a DM by a random walk, $x(t) = \sum_{n \in 1:t} \delta x_n$, where each of the steps $\delta x_n \sim (k\sin(h)\Delta, \Delta)$, called the momentary evidence, are normally distributed with mean $k\sin(h)\Delta$ and variance $\Delta$ (1:$t$ denotes the set of all steps up to time $t$). This representation is exact in the sense that it recovers the diffusion model, $\dot{x} = k\sin(h) + \eta(t)$, in the limit of $\Delta \to 0$.

For the standard diffusion model, the posterior probability of $\sin(h)$ after observing the stimulus for $t$ seconds, and under the assumption of a uniform prior, is given by Bayes rule

$$p\left(\sin(h) \mid \delta x_{1:t}\right) \propto \prod_{n \in 1:t} p\left(\delta x_n \mid \sin(h)\right) \propto N\left(\sin(h) \mid \frac{x(t)}{kt}, \frac{1}{k^2 t}\right), \tag{4}$$

where $\delta x_{1:t}$ is the momentary evidence up to time $t$. From this we can derive the belief that heading is rightward, resulting in

$$p\left(h > 0 \mid \delta x_{1:n}\right) = p\left(\sin(h) > 0 \mid \delta x_{1:t}\right) = \int_0^{\pi} p\left(\sin(h) \mid \delta x_{1:t}\right) dh = \Phi\left(\frac{x(t)}{\sqrt{t}}\right), \tag{5}$$

where $\Phi(\cdot)$ denotes the standard cumulative Gaussian function. This shows that both the posterior of the actual heading angle, as well as the belief about 'rightward' being the correct choice, only depend on $x(t)$ rather than the whole trajectory $\delta x_{1:t}$.

The above formulation assumes that evidence is constant over time, which is not the case for our stimuli. Considering the visual cue and assuming that its associated sensitivity varies with velocity $v(t)$, the momentary evidence $\delta x_{vis,n} \sim N\left(v_n k_{vis}(c)\sin(h)\Delta, \Delta\right)$ is Gaussian with mean $v_n k_{vis}(c)\sin(h)\Delta$, where $v_n$ is the velocity at time step $n$, and variance $\Delta$. Using Bayes rule again to find the posterior of $\sin(h)$, it is easy to shown that $x_{vis}(t)$ is no longer sufficient to determine the posterior distribution. Rather, we need to perform a velocity-weighted accumulation, $X_{vis}(t) = \sum_{n \in 1:t} v_n \delta x_{vis,n}$ to replace $x_{vis}(t)$, and replace time $t$ with $V(t) = \sum_{n \in 1:t} v_n^2 \Delta$, resulting in the following expression for the posterior

$$p\left(\sin(h) \mid \delta x_{vis,1:t}\right) = p\left(\sin(h) \mid X_{vis}(t), V(t)\right) = N\left(\sin(h) \mid \frac{X_{vis}(t)}{k_{vis}(c)V(t)}, \frac{1}{k_{vis}^2(c)V(t)}\right). \tag{6}$$

Consequently, the belief about 'rightward' being correct can also be fully expressed by $X_{vis}(t)$ and $V(t)$. This shows that optimal accumulation of evidence with a single-particle diffusion model with time-varying evidence sensitivity requires the momentary evidence to be weighted by its momentary sensitivity. A similar formulation holds for the posterior over heading based on the vestibular cue, however the vestibular cue is assumed to be weighted by the temporal profile of stimulus acceleration, instead of velocity.

When combining multiple cues into a single DM, $\dot{X}_{comb} = d(t)\left(d(t)k_{comb}\sin(h) + \eta_{comb}(t)\right)$, we aim to find expressions for $k_{comb}$ and $d(t)$ that keep the posterior over $\sin(h)$ unchanged, that is

$$p\left(\sin(h) \mid \delta x_{comb,1:t}\right) = p\left(\sin(h) \mid \delta x_{vis,1:t}, \delta x_{vest,1:t}\right). \tag{7}$$

$\delta x_{comb,1:t}$ is the sequence of momentary evidence in the combined condition, following $\delta x_{comb,n} \sim N\left(d_n k_{comb}(c)\sin(h)\Delta, \Delta\right)$. Expanding the probabilities reveals the equality to hold if the combined sensitivity is given by $k_{comb}^2(c) = k_{vis}^2(c) + k_{vest}^2$, and $d(t)$ is expressed by *Equation 3*, leading to *Equation 1* for optimally combining the momentary evidence (see *Supplementary file 1* for derivation).

## Model fitting

The model used to fit the behavioral data is described in the main text. We never averaged data across subjects as they feature qualitatively different behavior, due to different speed-accuracy tradeoffs. Furthermore, for subjects performing both the three-coherence and the six-coherence version of the experiment, we treated either version as a separate data set. For each modality/coherence combination (7 combinations for 3 coherences, 13 combinations for 6 coherences) we fitted one bias parameter that prevents behavioral biases from influencing model fits. The fact that performance of subjects often fails to reach 100% correct even for the highest coherences and largest heading angles was modeled by a lapse rate, which describes the frequency with which the subject makes a random choice rather than one based on accumulated evidence. This lapse rate was assumed to be independent of

stimulus modality or coherence, and so a single lapse rate parameter is shared among all modality/coherence combinations.

All model fits sought to find the model parameters $\varphi$ that maximize the likelihood of the observed choices and reaction times for each dataset. As in *Palmer et al. (2005)*, we have assumed the likelihood of the choices to follow a binomial distribution, and the reaction times of correct and incorrect choices to follow different Gaussian distributions centered on the empirical means and spread according to the standard error. Model predictions for choice fractions and reaction times for correct and incorrect choices were computed from the solution to integral equations describing first-passage times of bounded diffusion processes (*Smith, 2000*). See *Supplementary file 1* for the exact form of the likelihood function that was used.

To avoid getting trapped in local maxima of this likelihood, we utilized a three-step maximization procedure. First, we found a (possibly local) maximum by pseudo-gradient ascent on the likelihood function. Starting from this maximum, we used a Markov Chain Monte Carlo procedure to draw 44,000 samples from the parameter posterior under the assumption of a uniform, bounded prior. After this, we used the highest-likelihood sample, which is expected to be close to the mode of this posterior, as a starting point to find the posterior mode by pseudo-gradient ascent. The resulting parameter vector is taken as the maximum-likelihood estimate. All pseudo-gradient ascent maximizations were performed with the Optimization Toolbox of Matlab R2013a (Mathworks), using stringent stopping criteria (TolFun = TolX = $10^{-20}$) to prevent premature convergence.

## Additional information

### Competing interests

DEA: Reviewing editor, *eLife*. The other authors declare that no competing interests exist.

### Funding

| Funder | Grant reference number | Author |
| --- | --- | --- |
| National Institutes of Health | R01 DC007620 | Dora E Angelaki |
| National Institutes of Health | R01 EY016178 | Gregory C DeAngelis |
| National Science Foundation | BCS0446730 | Alexandre Pouget |
| U.S. Army Research Laboratory | Multidisciplinary University Research Initiative, N00014-07-1-0937 | Alexandre Pouget |
| Air Force Office of Scientific Research | FA9550-10-1-0336 | Alexandre Pouget |
| James S. McDonnell Foundation | | Alexandre Pouget |

The funders had no role in study design, data collection and interpretation, or the decision to submit the work for publication.

### Author contributions

JD, Conception and design, Analysis and interpretation of data, Drafting or revising the article; GCDA, AP, Conception and design, Drafting or revising the article; EMK, Acquisition of data, Drafting or revising the article; DEA, Conception and design, Acquisition of data, Drafting or revising the article

### Ethics

Human subjects: Informed consent was obtained from all participants and all procedures were reviewed and approved by the Washington University Office of Human Research Protections (OHRP), Institutional Review Board (IRB; IRB ID# 201109183). Consent to publish was not obtained in writing, as it was not required by the IRB, but all subjects were recruited for this purpose and approved verbally. Of the initial seven subjects, three participated in a follow-up experiment roughly 2 years after the initial data collection. Procedures for the follow-up experiment were approved by the Institutional Review Board for Human Subject Research for Baylor College of Medicine and Affiliated Hospitals (BCM IRB, ID# H-29411) and informed consent and consent to publish was given again by all three subjects.

# Additional files

**Supplementary files**

• Supplementary file 1. Detailed model derivation and description.

• Supplementary file 2. Outcome of additional statistical hypothesis tests.

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
