## [Decision Letter]

Thank you for sending your work entitled “Optimal multisensory decision-making in a reaction-time task” for consideration at *eLife.* Your article has been favorably evaluated by Eve Marder (Senior editor) and 2 reviewers, one of whom, Emilio Salinas, has agreed to reveal his identity.

The Senior editor and the two reviewers discussed their comments before we reached this decision, and the Senior editor has assembled the following comments to help you prepare a revised submission.

The authors carry out a detailed theoretical analysis of a vestibular-visual cue integration task, in which subjects can make a response at any time after the stimulus comes on. Unlike tasks that have fixed information delivery times, the behavioral thresholds in the combined task, in the present study, are not better than both of the individual thresholds. The reason for this is that subjects terminate evidence accumulation more quickly in the combined condition. The authors develop a model which incorporates time-varying evidence across both cues up to the reaction time (minus baseline stimulus-response processing) and they show that this model accurately characterizes reaction times and accuracy. They also show that the subjects approximately optimally integrate evidence.

This study represents both theoretical and empirical advances. The behavior has been carefully carried out, the data analysis is detailed and thorough, and the modeling provides and important insight into the behavioral process. I think both the experimental data and the modeling insights are quite compelling and novel. On one hand, multisensory experiments have become quite popular, and performance improvements have been amply documented both in terms of reaction times and of response accuracy. But in retrospect it seems rather surprising that multisensory enhancement has not been studied for the more natural, simultaneous condition in which both time and accuracy are in play. This work not only fills in this gap, but also presents results that may seem quite paradoxical when RT and accuracy are analyzed separately from each other. Surprisingly, the combined condition does not produce better (i.e., more accurate) performance, as one may have thought based on previous results, but mostly faster performance.

The study also presents fits of the experimental data to a generalized version of the diffusion model that works with two independent streams of sensory evidence. The model may not be the ultimate one – it is rather abstract and provides little mechanistic intuition about the underlying neuronal coding schemes and circuit interactions – but it does serve its purpose at this point, which is to prove a quantitative statistical benchmark for measuring the effectiveness of those underlying neural interactions, as well as a framework for testing and generating hypotheses. The generalization to two evidence streams, rather than one, and to a time-dependent reliability of the sensory evidence is a clever and useful theoretical advance, and it describes the data quite well.

Minor comments:

1) How were the degrees of freedom calculated for the BIC? Was the model probability calculated by computing a BIC for each subject, and then summing these across subjects? Another approach that is used in functional imaging is to compute exceedance probabilities. Model evidence across multiple subjects inflates degrees of freedom, and exceedance probabilities have been developed to deal with that problem. This is similar to fixed effects vs. mixed effects (or hierarchical) models for analyzing behavioral data across multiple subjects.

2) Within the Results section there is a paragraph about how to set the noise terms in the model, but the reader finds that out several lines ahead. This would be easier to follow if an introductory sentence were added along the lines of 'The noise terms eta_vis and eta_vest play crucial roles in the model, as they relate to the reliability of the momentary sensory evidence. To specify the manner in which such noise may depend on motion coherence, we relied on fundamental assumptions about how optic flow stimuli are represented by the brain...”

---

## [Author Response]

*1) How were the degrees of freedom calculated for the BIC? Was the model probability calculated by computing a BIC for each subject, and then summing these across subjects? Another approach that is used in functional imaging is to compute exceedance probabilities. Model evidence across multiple subjects inflates degrees of freedom, and exceedance probabilities have been developed to deal with that problem. This is similar to fixed effects vs. mixed effects (or hierarchical) models for analyzing behavioral data across multiple subjects*.

As we fitted the model parameters for each subject separately, the BIC was computed for each subject separately and then summed. The details of this procedure are described in the Data Analysis subsection in Methods. All but one model discussed in the main text have the same number of parameters, such that other approaches to taking the number of model parameters into account would have led to the same result.

As suggested by the reviewers, we have additionally added a random-effects Bayesian model comparison, which was added to Figure 7—figure supplement 1 (panels b and c). The results of this analysis are consistent with our previous BIC analysis, adding strength to the conclusions. We thank the reviewers for this good suggestion.

*2) Within the Results section there is a paragraph about how to set the noise terms in the model, but the reader finds that out several lines ahead. This would be easier to follow if an introductory sentence were added along the lines of 'The noise terms eta_vis and eta_vest play crucial roles in the model, as they relate to the reliability of the momentary sensory evidence*. *To specify the manner in which such noise may depend on motion coherence, we relied on fundamental assumptions about how optic flow stimuli are represented by the brain...”*

Thank you for this suggestion. We have modified the beginning of this paragraph as suggested.